# The dependency structure of international commodity and stock markets after the Russia-Ukraine war

**Cheng Zhang[1], Shuo Liu[2], Mimi Qin[1], Bin Gao[1] ***

**1** School of Economics, Guangxi Minzu University, Nanning, China, **2** School of Economics, Capital University of Economics and Business, Beijing, China

\* gaobin@gxmzu.edu.cn

## Abstract

In recent years, the international community has witnessed many crisis events, and the Russia-Ukraine war, which broke out on 24th February 2022, has increased international policy uncertainty and impacted the current world commodity and financial markets. Thus, we try to capture how the Russia-Ukraine war has affected the correlation structure of international commodity and stock markets. We study six groups of commodity daily returns and one group of stock daily returns and select the sample from 24th February 2022 to 1st June 2022 as the sample during the Russia-Ukraine war; in addition, we select the sample from 1st December 2019 to 31st December 2020 as the sample during COVID-19 control group, and the sample from 1st January 2014 to 31st December 2017 as the non-extreme event control group, to explore the correlation structure of international commodity and stock markets before the war, and to compare and uncover the impact of the uncertain event of the Russia-Ukraine war on the commodity and stock markets. In this paper, the marginal density function of each series is constructed using the ARMA-GARCH-std method, and the R-Vine copula model is built based on the marginal density function to analyze the correlation relationship between each market. From the Tree1 of the Vine copula, it is found that crude oil becomes the core connecting each commodity market and the stock market during the Russia-Ukraine war. The price fluctuations of crude oil may be contagious to agricultural and precious metal markets in the same direction, while the stock market price fluctuations are inversely correlated with commodity markets. Comparison with the selected control group sample reveals that the Russia-Ukraine war increases the correlation between the markets and enhances the possibility of risk transmission. The core of the correlation structure shifts from agricultural commodities and precious metals to crude oil after the Russia-Ukraine war.

## Introduction

In recent decades, the deepening integration of global trade has provided investors in financial markets with a broader and more complex range of investment choices and portfolio strategies, while the trend of commodity financialization has also intensified [1]. The link between

**Data Availability Statement:** The data are all contained within the manuscript.

**Funding:** 'Guangxi One Thousand Young and Middle-Aged College and University Backbone

Teachers Cultivation Program' Humanities and social sciences projects (2020QGRW016).

**Competing interests:** The authors have declared that no competing interests exist.

international commodity markets and financial markets, represented by stock markets, has become increasingly closer, and the price fluctuations of assets across these markets are often interrelated. Baker et al. [2] highlight that in the current context of highly interconnected global financial and commodity markets, exogenous shocks such as the COVID-19 pandemic, which emerged at the end of 2019, significantly exacerbate market volatility. This outbreak not only precipitated a global public health crisis but also resulted in heightened economic uncertainty and a macroeconomic recession. Subsequently, the Russo-Ukrainian War, which began on February 24, 2022, introduced further substantial uncertainty into the global policy environment, exacerbating volatility in international financial markets and potentially having enduring effects on the global economy [3,4]. Policy uncertainty alters investment decisions and consumer behavior, thereby affecting financial market returns and introducing spillover effects that increase market volatility [5]. Albulescu et al. [6] noted that policy uncertainty amplifies the potential for risk transmission between commodity markets and financial markets. As a result, studies investigating the relationship between commodity markets and financial markets under external shocks have emerged, covering markets such as oil, gold, stock markets, and exchange rates [7,8]. Brune et al. [9] specifically pointed out that the sudden outbreak of war increases market uncertainty, and international financial markets often react negatively to war and armed conflict events, which is reflected in falling stock prices.

The Russia-Ukraine war, as a sudden geopolitical conflict, brought new policy uncertainty to the international community [10]. *The New York Times* reported that on the day the war broke out, the S&P 500 index fell by more than 10%. This suggests that the Russia-Ukraine war may affect international stock markets and spill over into commodity markets, thereby altering the existing relationship between international financial markets and commodity markets. The Russia-Ukraine war not only increased the uncertainty surrounding business operations but also had a significant negative impact on global stock market returns. Although the war occurred primarily between Russia and Ukraine, economic globalization magnified its impact, causing ripple effects on the neighboring countries and the UN member states calling for a ceasefire[10]. In the commodity market, the war is expected to have a particularly pronounced impact due to Russia's key role in energy supply and Ukraine's importance in agricultural supply, with oil being a critical commodity in international markets [11].

With the deepening of global economic integration, the interdependence between international financial markets and commodity markets has grown, meaning that price fluctuations and risks in one asset market may spill over to other markets. Analyzing market dependencies during major external shocks helps investors better allocate assets. While there has been existing research on the relationship between commodity and financial markets under external shocks, studies on the deep structural changes in dependencies induced by the Russia-Ukraine war are still limited. This study aims to analyze the relationship between international commodity markets and stock markets under the external shock of the Russia-Ukraine war. We selected the prices of six commodity futures (oil, gold, silver, soybean, corn, and wheat) and the S&P 500 index, using ARMA-GARCH models to fit the marginal distributions of each market and employing the R-Vine Copula structure to analyze the dependencies between these marginal distributions. To ensure the robustness of the data, we excluded irregular trading days and constructed independent R-Vine Copula structures for the periods of the Russia-Ukraine war, the pre-war COVID-19 pandemic, and periods without major external shocks. This analysis aims to better understand the impact of the Russia-Ukraine war on the dependence structure between commodity and stock markets.

By comparing the dependency structures between commodity and stock markets before and after the war, as well as during the pandemic, we found that the Russia-Ukraine war significantly strengthened the dependence between commodity and stock markets, with oil

emerging as a core linked commodity, showing a pronounced transmission effect. Additionally, the inverse relationship between the stock market and precious metals indicates that investors tend to treat precious metals as safe-haven assets in high-risk environments. Furthermore, the price fluctuations of agricultural commodities during the Russia-Ukraine war showed stronger correlations with the energy market, further demonstrating the heightened spillover effects between markets under the war context. These findings align with our initial expectations.

The remainder of the paper is structured as follows: The Methodology section introduces the ARMA-GARCH model and R-Vine Copula method; The Empirical Studies and Analysis section describes the data sources and sample characteristics, providing descriptive statistics; The Modeling Multiple Dependence Structures section presents the empirical analysis results, exploring the differences in market dependencies under different backgrounds; The Conclusion section concludes the paper, discussing the implications of these findings for portfolio design and risk management.

## Methodology

### Marginal distribution modeling

**ARMA.**  The autoregressive moving average model (ARMA) is a model consisting of a combination of the autoregressive model (AR) and moving average model (MA) in the general form of ARMA(p,q), which is a p-order autoregressive (AR) q-order moving average (MA) model. In the ARMA(p,q) model, the current value in the series is related to the previous fluctuations. Combining the expressions of the AR(p) and MA(q) models, the ARMA(p,q) model is expressed as:

$$x_t = c + \varepsilon_t + \sum\nolimits_{i=1}^{p} \varphi_i \, x_{t-i} + \sum\nolimits_{j=1}^{q} \theta_j \, \varepsilon_{t-j} \tag{1}$$

Where $\{x_t\}$ denotes the desired log-return series and $\{\varepsilon_t\}$ denotes the residual series; where $x_{t-i}$ denotes the log-return with lag i and $\varepsilon_{t-j}$ is the residual with lag j. p is the optimal order of the AR model, and q is the optimal order of the MA model. $\varphi_i$ and $\theta_j$ represent the coefficients corresponding to the respective variables.

### GARCH

Analysis of real-life financial return time series data often reveals that these series are characterized by spikes, thick tails, and autocorrelation, and there is a clustering effect on their fluctuations [12]. The autoregressive conditional heteroskedasticity model (ARCH) method, first proposed by Engle [13], is often used to solve the problem of the volatility clustering effect of financial time series and makes an important contribution to mastering the volatility problem of financial time series. However, in the process of practical application, ARCH may have problems with high lag order q and reduced efficiency of parameter estimation, which in turn will lead to poor fitting of the ARCH model. Therefore, on this basis, the generalized autoregressive conditional heteroskedasticity model (GARCH) model proposed by Bollerslev [14] concerning the idea of the ARMA model, which uses lagged terms with less conditional variance instead of ARCH lagged terms, can be used as an alternative model of higher-order ARCH to better correct the heteroskedasticity problem long-term autocorrelation problem. Compared with ARCH models, GARCH can better analyze the volatility of financial time series with long-term heteroskedasticity autocorrelation problems. The common types of GARCH class models are sGARCH, eGARCH, tGARCH, GJR-GARCH, and APARCH, according to the distinction of distribution characteristics. According to the AIC, BIC, and

HQ criteria, the models selected in this paper include sGARCH, APARCH, and GJR-GARCH. Among them, AIC (Akaike Information Criterion), BIC (Bayesian Information Criterion), and HQ (Hannan-Quinn Information Criterion) are standards used for model selection, helping to determine the optimal model specification. Both AIC and BIC balance the model's fit and complexity; AIC tends to select models with higher complexity, while BIC favors more parsimonious models. The HQ criterion provides a compromise between AIC and BIC. Generally, smaller AIC, BIC, and HQ values indicate that the model has a better fit and lower complexity.

## ARMA-GARCH

To better eliminate the autocorrelation and heteroskedasticity of financial market time series, In this paper, we choose to use the ARMA(p,q)-GARCH(1,1) model to simulate the marginal distribution functions of the commodity and stock markets as a preliminary preparation for analyzing the interdependence structure of each market. Next, we construct the ARMA(p,q)-GARCH(1,1) model using the following equation.

$$\varepsilon_t = \sqrt{h_t} e_t \tag{2}$$

$$h_t = \alpha_0 + \alpha_1 \varepsilon_{t-1}^2 + \beta_1 h_{t-1} \tag{3}$$

$$e_t \sim N(0,1) \tag{4}$$

Eq (2) is the ARMA process, $\varepsilon_t$ and $e_t$ denote the residuals and standardized residuals, respectively, and Eq (3) is the mean equation process, where $\alpha_1$ and $\beta_1$ are both greater than 0 and the sum is less than 1. Eq (4) assumes that the normalized residual distribution follows a normal distribution.

## R-Vine copula approach

This paper aims to capture the impact of the Russia-Ukraine war on the dependency structure of the international commodity and stock markets; therefore, we use the R-Vine copula approach to explore the degree of dependency between the markets and the direction of the dependency.

Referring to existing studies, it is known that the copula method was first proposed by Sklar [15], and Sklar considers that any n-dimensional joint distribution function can be split into n marginal distribution functions. A copula function is defined by the existence of a copula function such that the following equation holds.

$$F(x_1, x_2, \ldots, x_n) = C(F(x_1), F(x_2), \cdots, F(x_n)) \tag{5}$$

Where $F(x_1, x_2, \ldots, x_n)$ is the joint distribution function of each unitary density function $F(x_1)$, $F(x_2), \cdots, F(x_n)$, and $C(\cdot, \cdots, \cdot)$ is the corresponding n-dimensional copula function. This demonstrates that the marginal and joint distributions of multivariate random variables can be connected by constructing a copula function. Copula provides a modeling method for analyzing the interdependence among financial markets.

Building on this, Joe [16] proposed the Pair-Copula Construction (PPC). Subsequently, Bedford and Cooke [17] developed the R-Vine Copula construction based on PPC. The R-Vine Copula can decompose a multivariate density function into marginal densities and a series of unconditional or conditional pair-Copulas, thereby capturing the dependence and degree of dependence between any two variables in an n-dimensional space. The definition of

the R-Vine Copula is as follows:

$$f(x_1, x_2, \ldots, x_n) = \prod_{i=1}^{n} f(x_i) \prod_{i=1}^{n-1} \prod_{e \in E_i} c_{j_e, k_e | d_e} \{F(j_e | d_e), F(k_e | d_e); d_e\} \qquad (6)$$

$f(x_i)$ is the edge density of each financial market return $\{x_i\}$, and $c_{j_e, k_e | d_e}$ denotes the pair-copula density function corresponding to the edge $e$ connecting variable $j$ and variable $k$ in the $i$th tree. $d_e$ represents the set of conditions, which is the intersection of the two sets of nodes reachable by variables $j$ and $k$ in the first tree. $E_i = \{E_1, E_2, \ldots, E_{n-1}\}$ is the set of edges in the $i$th tree.

The R-Vine Copula structure is generated according to the maximum spanning tree algorithm and consists of three necessary components: tree, edges, and nodes [18]. An n-dimensional R-Vine Copula model contains n-1 layers of trees, each with a different structure, where the first layer of trees represents the unconditional coupling of variables with the strongest dependencies. Each layer of the tree contains a series of edges and nodes, and each node represents a variable, while the edges represent each edge represents the degree of dependence between two variables. Each edge can be measured by the absolute value of Kendall's tau to measure the degree of dependence. Specifically, we can seek the optimal edge in the R-Vine copula structure by the following equation.

$$\max \sum_{\substack{\text{edges} \\ e_{ij} \in \text{ in spanning tree}}} |\tau_{ij}|, i \neq j \qquad (7)$$

$\tau_{ij}$ is the Kendall rank correlation coefficient tau between nodes $i$ and $j$.

After constructing the R-Vine copula structure, we refer to the use of the AIC criteria to select the optimal pair-copula function [19].

## Empirical studies and analysis

### Data

The data for this study is sourced from the Nanhua Commodity Index Database, covering key financial indicators: WTI crude oil futures settlement prices, the closing prices of gold on the London International Financial Futures Exchange (LIFFE), closing prices of silver in London, and closing prices of soybean, corn, and wheat futures on the Chicago Board of Trade (CBOT), along with the S&P 500 index. The selection of commodity and stock price indexes is shown in Table 1. The research period is specifically chosen from February 24th, 2022, to June 1st, 2022, corresponding to the Russia-Ukraine conflict period. For the control group, data from December 1st, 2019, to December 31st, 2020, is selected to represent the COVID-19 period, and data from January 1st, 2014, to December 31st, 2017, is selected as the sample for a period without extreme events. In this paper, we meticulously filtered the sample data,

**Table 1. Commodity and stock price index selection.**

| Commodity and Stock Indexes | Code | Index Source |
|---|---|---|
| Crude Oil Futures | OIL | WTI Crude Oil Futures Settlement Price |
| Gold Price | AU | LIFFE London Gold Closing Price |
| Silver Price | AG | LIFFE London Silver Closing Price |
| Bean Futures Prices | BEAN | CBOT Soybean Futures Closing Price |
| Corn Futures Price | CORN | CBOT Corn Futures Closing Price |
| Wheat Futures Price | WHEAT | CBOT Wheat Futures Closing Price |
| Stock Price Index | S&P500 | Standard & Poor's 500 Index |

removing non-synchronous trading dates and excluding records with missing values. Applying Eq (8) to the seven data sets, we derived the daily return series, which exhibit temporal smoothness and are regarded as first-order differenced series.

$$r_{i,t} = \ln(P_{i,t}) - \ln(P_{i,t-1}) \tag{8}$$

where $r_{i,t}$ represents the daily return in period $t$, $P_{i,t}$ is the closing price of $i$-series in period $t$, and $P_{i,t-1}$ is the closing price of $i$-series in period $t$-$1$.

After the above processing, this paper obtains 67 sets of samples during the Russia-Ukraine war. Meanwhile, to compare and derive the changes in the impact of this war on the commodity and stock market interdependence structure and to ensure the robustness of the empirical model constructed in this paper, two control groups are selected: the pre-war COVID-19 pandemic period and the period without extreme risk events. The pre-war COVID-19 period is defined as 1st December 2019 to 31st December 2020, with a total of 269 sample groups. The no extreme risk event period is defined as 1st January 2014 to 31st December 2017, during which the European debt crisis and China-US trade issues did not occur, with a total of 491 sets of samples. Due to the limitation of article length, this paper mainly presents the test results for the samples during the Russia-Ukraine war period and the samples during the COVID-19 epidemic, and the statistical tests and the ARMA-GARCH process for the control group without extreme events are no longer placed in the paper.

## Descriptive statistics

Descriptive statistics for seven sets of yield series for the Russia-Ukraine war period and the control COVID-19 pandemic period are shown in Table 2. Due to the word limit of the paper, the test results and the ARMA-GARCH process for the control group are not placed in the paper.

First, observing the sample during the Russia-Ukraine war shows that the spike thick tail phenomenon is no longer obvious, and the Jarque-Bera test shows that the AU, BEAN, CORN, and WHEAT series reject the original hypothesis of normal distribution at different significant levels, and the overall view reflects skewed distribution. In addition, each series also significantly rejects the original hypothesis of the ADF test, and the series is smooth. In addition, except for the OIL and BEAN series, the ARCH effect is also present in each series. And there is also an autocorrelation effect for each series except the BEAN series.

Second, observing the kurtosis and skewness of the samples during the COVID-19 epidemic period shows that all seven groups of return series reveal the characteristics of spikes and thick tails, and the Jarque-Bera test shows that each series rejects the original hypothesis of normal distribution at different significant levels, reflecting the skewed distribution of spikes and thick tails. In addition, further statistics on the return series show that all seven sets of series reject the original hypothesis at the 1% significant level in the ADF test, and the series are all smooth. The ARCH-LM test with a lag of 15 periods is significant, and there is an ARCH effect for each series. In addition, observing the lagged 20th order statistic Q (20) of the Ljung-Box test reveals that each series rejects the original hypothesis, and there is autocorrelation.

## Marginal distribution models: ARMA-GARCH

Firstly, to address the issues of autocorrelation and heteroskedasticity in time series data, we have employed the ARMA-GARCH modeling framework. Specifically, we have fitted an ARMA(m, n)-GARCH(1,1) model to each series. The optimal orders of the ARMA model, denoted by m and n, were determined using the auto.arima function in R, which selects the

**Table 2. Summary statistics of the observed returns.**

| | OIL | AU | AG | BEAN | CORN | WHEAT | S&P500 |
|---|---|---|---|---|---|---|---|
| | I During the Russia-Ukraine War | | | | | | |
| N | 67 | 67 | 67 | 67 | 67 | 67 | 67 |
| Min | -0.106 | -0.028 | -0.043 | -0.077 | -0.098 | -0.165 | -0.043 |
| Max | 0.165 | 0.059 | 0.060 | 0.043 | 0.054 | 0.137 | 0.066 |
| Mean | 0.000 | 0.000 | 0.000 | 0.000 | -0.001 | -0.001 | 0.001 |
| Std | 0.054 | 0.016 | 0.024 | 0.021 | 0.029 | 0.048 | 0.025 |
| Skewness | 0.210 | 0.776 | 0.401 | -0.380 | -0.584 | -0.535 | 0.465 |
| Kurtosis | -0.051 | 1.163 | -0.407 | 1.032 | 0.647 | 1.666 | -0.356 |
| J-B test | 0.518 | 11.681** | 2.182 | 5.413* | 5.590* | 12.478*** | 2.740 |
| ADF test | -4.777*** | -5.228*** | -5.064*** | -5.330*** | -5.590*** | -5.083*** | -4.821*** |
| ARCH-LM | 8.581 | 23.094* | 11.701** | 9.046 | 15.341*** | 21.525*** | 11.467** |
| Q (20) | 42.275*** | 48.128*** | 47.283*** | 18.444 | 41.532*** | 44.431*** | 56.567*** |
| | II During the COVID-19 | | | | | | |
| N | 269 | 269 | 269 | 269 | 269 | 269 | 269 |
| Min | -92.131 | -5.920 | -9.434 | -4.438 | -6.411 | -5.500 | -18.875 |
| Max | 51.773 | 5.566 | 10.841 | 4.395 | 4.871 | 7.817 | 21.646 |
| Mean | -0.002 | 0.003 | 0.002 | 0.001 | -0.007 | -0.021 | -0.003 |
| Std | 9.850 | 1.568 | 2.967 | 1.261 | 1.761 | 2.277 | 3.518 |
| Skewness | -2.938 | 0.242 | 0.110 | -0.039 | -0.047 | 0.471 | -0.289 |
| Kurtosis | 33.196 | 2.006 | 1.276 | 0.967 | 0.927 | 0.423 | 11.358 |
| J-B test | 12945*** | 49.471*** | 19.707*** | 11.192** | 10.356* | 12.296*** | 1477.1*** |
| ADF test | -11.189*** | -11.024*** | -11.181*** | -9.490*** | -9.845*** | -10.102*** | -10.686*** |
| ARCH-LM | 34.209*** | 59.594*** | 31.184*** | 45.059*** | 43.035*** | 42.266*** | 165.84*** |
| Q (20) | 118.710*** | 95.164*** | 92.217*** | 85.293*** | 83.302*** | 123.890*** | 463.690*** |

Note

*, **, *** denote significant at the 10%, 5%, and 1% levels. Q (20) is the lagged 20th order statistic in Ljung-Box, and the lagged order of the ARCH-LM test is 5 in I. The lagged order of the ARCH-LM test is 15 in period II.

best orders based on the Akaike Information Criterion (AIC) and the Bayesian Information Criterion (BIC). Following the selection of the optimal lag orders, we conducted parameter estimation, and the results are presented in Table 3. And Table 3 includes the estimated parameters for both the ARMA and GARCH components, and we assessed the model fit by examining the significance of the coefficients and the residuals. In this way, we ensure that the model accurately captures the dynamic characteristics of each series.

Next, in selecting the type of GARCH model, we considered sGARCH, APARCH, and GJR-GARCH models. The optimal model was chosen using the AIC, BIC, and HQ criteria to achieve a balance between model fit and complexity, thereby avoiding overfitting. After fitting the model, parameter estimation was conducted, as presented in Table 4, where most parameters are statistically significant.

Then, regarding the choice of distribution, we base our selection on the Jarque-Bera (J-B) test results in Table 4, which indicate whether the data conforms to a normal distribution. For series with non-significant J-B test results, we use the normal distribution (norm); for those significantly deviating from normality, we select the Student's t-distribution (std) or the Generalized Error Distribution (ged). The normal distribution is appropriate for data exhibiting relatively symmetric returns with no substantial skewness. In contrast, the t-distribution suits series with fat tails, commonly observed in financial data, while the ged is preferable when

**Table 3. ARMA parameter estimation.**

| Parameters | Estimation of ARIMA parameters during the Russia-Ukraine war | | | | | | |
|---|---|---|---|---|---|---|---|
| | OIL | AU | AG | BEAN | CORN | WHEAT | S&P500 |
| ar(1) | -0.691*** | -0.928*** | -0.943*** | -0.836*** | -0.304** | -0.537*** | -0.923*** |
| ar(2) | -0.528*** | -0.678*** | -0.621*** | -0.822*** | | | -0.718*** |
| ar(3) | -0.394*** | -0.427*** | -0.246** | -0.437*** | | | -0.564*** |
| ar(4) | | -0.202** | | -0.289** | | | -0.204* |
| ar(5) | | | | | | | |
| ma(1) | | | | | -1.000*** | | |
| ma(2) | | | | | | | |
| intercept | 0.000 | 0.000 | 0.001 | 0.000 | 0.000 | 0.000 | 0.000 |
| AIC | -222.299 | -391.098 | -342.646 | -350.204 | -336.676 | -229.126 | -335.615 |
| BIC | -211.276 | -377.870 | -331.623 | -336.976 | -327.857 | -222.512 | -322.387 |
| HQ | -6.147 | -8.656 | -7.945 | -8.049 | -7.926 | -6.267 | -7.830 |
| LL | 116.150 | 201.549 | 176.323 | 181.102 | 172.338 | 117.563 | 173.808 |
| Parameters | Estimation of ARIMA parameters during COVID-19 | | | | | | |
| | OIL | AU | AG | BEAN | CORN | WHEAT | S&P500 |
| ar(1) | -0.933*** | -0.610*** | -0.484*** | | 0.051 | -0.552 | -0.718*** |
| ar(2) | -0.885*** | -0.268*** | | | -0.067 | | |
| ar(3) | -0.789*** | | | | 0.032 | | |
| ar(4) | -0.391*** | | | | -0.107 | | |
| ar(5) | -0.160*** | | | | | | |
| ma(1) | | | | -0.855*** | -1.000 | | |
| ma(2) | | | | -0.145** | | | |
| intercept | 0.003 | 0.001 | 0.001 | -0.002** | -0.003 | -0.014 | 0.000 |
| AIC | 1818.344 | 921.556 | 1282.048 | 752.220 | 906.978 | 2238.286 | 2127.089 |
| BIC | 1843.507 | 935.935 | 1292.832 | 766.599 | 932.141 | 2251.053 | 2139.856 |
| HQ | 3.941 | 0.595 | 1.930 | -0.053 | 0.537 | 1.460 | 1.246 |
| LL | -902.170 | -456.780 | -638.020 | -372.110 | -446.490 | -553.960 | -586.470 |

Note

*, **, *** indicate significant at the 10%, 5%, and 1% levels.

there is evidence of high kurtosis or extreme values, as it better accommodates heavy tails than the normal distribution. These distribution choices are based on the specific characteristics of each series to ensure accurate representation of data properties.

Finally, following the marginal distribution fitting, we applied probability integral transformations to convert the distributions into standardized residuals, followed by the Kolmogorov-Smirnov (KS) test. The results of the KS test failed to reject the null hypothesis, suggesting that the transformed series follow a uniform distribution, thereby meeting the conditions required for constructing a Vine copula model. Consequently, the ARMA-GARCH model provides a robust foundation for constructing the Vine copula dependency structure. Due to length limitations, detailed KS test results are omitted.

## Modeling multiple dependence structures

In this section, we utilize the R programming environment to construct 7-dimensional Vine Copula models based on standardized residuals, aiming to estimate the dependency structures among international commodity and stock markets during both the Russia-Ukraine war and the control period. Vine Copula models allow us to capture complex dependency relationships

**Table 4. ARMA-GARCH-std parameter estimation.**

| | During the Russia-Ukraine War | | | | | | |
|---|---|---|---|---|---|---|---|
| | OIL | AU | AG | BEAN | CORN | WHEAT | S&P500 |
| GARCH model | GJRGARCH (1,1)-norm | APARCH (1,1)-std | sGARCH (1,1)-norm | sGARCH (1,1)-std | apARCH (1,1)-ged | sGARCH (1,1)-std | sGARCH (1,1)-norm |
| μ | 0.001*** | 0.000 | 0.000 | 0.000 | 0.001 | 0.000 | 0.000 |
| | 0.000 | 0.016 | 0.001 | 0.001 | 0.001 | 0.002 | 0.001 |
| ω | 0.000 | 0.000 | 0.000 | 0.000 | 0.000 | 0.000** | 0.000 |
| | 0.000 | 0.000 | 0.000 | 0.001 | 0.000 | 0.000 | 0.000 |
| α | 0.056*** | 0.021 | 0.000 | 0.000 | 0.371 | 0.095 | 0.095 |
| | 0.000 | 18.525 | 0.001 | 0.035 | 0.286 | 0.084 | 0.094 |
| β | 1.000*** | 0.806 | 0.994*** | 0.000 | 0.204 | 0.832*** | 0.831*** |
| | 0.000 | 36.993 | 0.003 | 0.823 | 0.501 | 0.062 | 0.066 |
| γ | -0.134*** | 0.555 | | | -0.124 | | |
| | 0.001 | 550.240 | | | 0.267 | | |
| δ | | 3.302 | | | 3.500*** | | |
| | | 63.713 | | | 0.242 | | |
| shape | | 11.679 | | 6.633* | 2.369*** | 7.767 | |
| | | 163.298 | | 3.991 | 0.722 | 5.403 | |
| LL | 120.279 | 203.576 | 176.698 | 179.693 | 163.492 | 122.480 | 173.017 |
| | During the COVID-19 | | | | | | |
| GARCH model | OIL | AU | AG | BEAN | CORN | WHEAT | S&P500 |
| | GJRGARCH (1,1)-ged | GJRGARCH (1,1)-std | sGARCH (1,1)-std | APARCH (1,1)-std | sGARCH (1,1)-std | sGARCH (1,1)-std | GJRGARCH (1,1)-ged |
| μ | 0.035*** | -0.029 | -0.054 | -0.003 | -0.001*** | -0.020 | 0.032* |
| | 0.002 | 0.025 | 0.046 | 0.000 | 0.001 | 0.036 | 0.019 |
| ω | 0.507*** | 0.119 | 0.865 | 0.002 | 0.053*** | 0.009 | 0.101*** |
| | 0.237 | 0.074 | 0.747 | 0.056 | 0.012 | 0.026 | 0.050 |
| α | 0.426*** | 0.149** | 0.211 *** | 0.000 | 0.000 | 0.004 | 0.474*** |
| | 0.116 | 0.069 | 0.092 | 0.068 | 0.000 | 0.008 | 0.126 |
| β | 0.744*** | 0.744*** | 0.736*** | 0.999*** | 0.972*** | 0.995*** | 0.700*** |
| | 0.052 | 0.095 | 0.129 | 0.003 | 0.005 | 0.001 | 0.055 |
| γ | -0.342*** | 0.151 | | -1.000 | | | -0.351** |
| | 0.132 | 0.197 | | 0.081 | | | 0.144 |
| δ | | | | 1.286 ** | | | |
| | | | | 0.591 | | | |
| shape | 0.974*** | 5.939*** | 3.981 *** | 7.288 | 4.204 *** | 11.964 | 1.414*** |
| | 0.135 | 2.064 | 1.218 | 19.958 | 1.283 | 9.552 | 0.163 |
| LL | -716.9853 | -430.634 | -616.4662 | -356.599 | -438.287 | -551.850 | -483.634 |

Note: The table reports Robust Standard Errors.

*, **, *** indicate significant at the 10%, 5%, and 1% levels.

by decomposing them into pairwise copulas, using Kendall's rank correlation coefficient (tau) to measure dependency strength and upper (utd) and lower tail dependence (ltd) coefficients to reflect dependency under extreme conditions. Referring to the maximum spanning tree principle from Diβmanna et al. [18], parameters are estimated using the maximum likelihood principle, while the optimal pair copula type is chosen based on the AIC.

The tree diagram shows the estimated dependence structure across commodity and stock markets during the study period, with assets labeled 1 through 7 representing "OIL," "AU,"

"AG," "BEAN," "CORN," "WHEAT," and "S&P500," respectively. To conserve space, each table in this section reports only the parameter estimates for the first level of Tree1.

### R-Vine copula during the Russia-Ukraine war

Observing Table 5, we can see that: During the Russia-Ukraine war, the dependency structures and price co-movements between commodity and stock markets exhibited significant characteristics, further underscoring the war's profound impact on global financial markets.

Firstly, from the perspective of dependency, aside from the negative dependency between gold and the stock market (with a dependency coefficient of -0.199), all other markets showed positive dependency coefficients, with Kendall's rank correlation $\tau$ ranging between 0.237 and 0.609. The correlation between gold and silver prices was the strongest, with a coefficient of 0.609. This positive correlation trend became more pronounced during the Russia-Ukraine war, indicating a heightened contagion effect in their prices compared to the COVID-19 period, with a stronger co-movement between the two assets. This suggests that the safe-haven properties of gold and silver were further reinforced in the context of the war, solidifying their interlinked price dynamics. Additionally, unlike the trend observed during the COVID-19 pandemic, gold and the stock market exhibited an inverse trend during the Russia-Ukraine war. This divergence provides new possibilities for asset allocation, indicating that investors could use gold as a hedge against stock market volatility, thereby enhancing portfolio stability.

Secondly, in terms of distribution characteristics, the dependency structures across markets displayed diversity: the relationship between crude oil and corn prices followed a t-Copula model, exhibiting symmetric tail dependence, meaning that these two assets could experience similar price movements under extreme market conditions (both upward and downward). The dependency structure between crude oil and silver, on the other hand, followed a 90-degree rotated Gumbel Copula (SG), displaying lower tail dependency (ltd) dependence, which suggests stronger co-movements during market downswings. The remaining market relationships were characterized by Gaussian copulas, indicating tail independence, where extreme price co-movements are typically absent. This diversity in dependency structures highlights the varying sensitivity to risk contagion among different commodity markets during wartime.

Fig 1 reports the structural characteristics of vines during the Russia-Ukraine war. From a structural perspective, the crude oil market emerged as a crucial link between the commodity and stock markets during the Russia-Ukraine war. The correlation coefficients of crude oil with silver, corn, and soybean were 0.244, 0.461, and 0.272, respectively, indicating that price fluctuations in the oil market could potentially transmit to agricultural and precious metal markets. This phenomenon aligns with global geopolitical developments. As a critical commodity behind the war, crude oil has become central to the global supply chain and price

**Table 5. Russia-Ukraine war R-Vine copula tree1 estimate results.**

| tree | Edge | family | cop | par | par2 | tau | utd | ltd |
|------|------|--------|------|--------|-------|--------|-------|-------|
| 1    | 6,5  | 2      | t    | 0.662  | 6.567 | 0.461  | 0.252 | 0.252 |
|      | 5,1  | 1      | N    | 0.426  | 0.000 | 0.280  | 0.000 | 0.000 |
|      | 1,4  | 14     | SG   | 1.373  | 0.000 | 0.272  | 0.000 | 0.343 |
|      | 1,3  | 4      | G    | 1.322  | 0.000 | 0.244  | 0.311 | 0.000 |
|      | 3,2  | 1      | N    | 0.817  | 0.000 | 0.609  | 0.000 | 0.000 |
|      | 2,7  | 34     | G270 | -1.249 | 0.000 | -0.199 | 0.000 | 0.000 |

Note: 1 to 7 represent "OIL," "AU," "AG," "BEAN," "CORN," "WHEAT," and "S&P500," respectively.

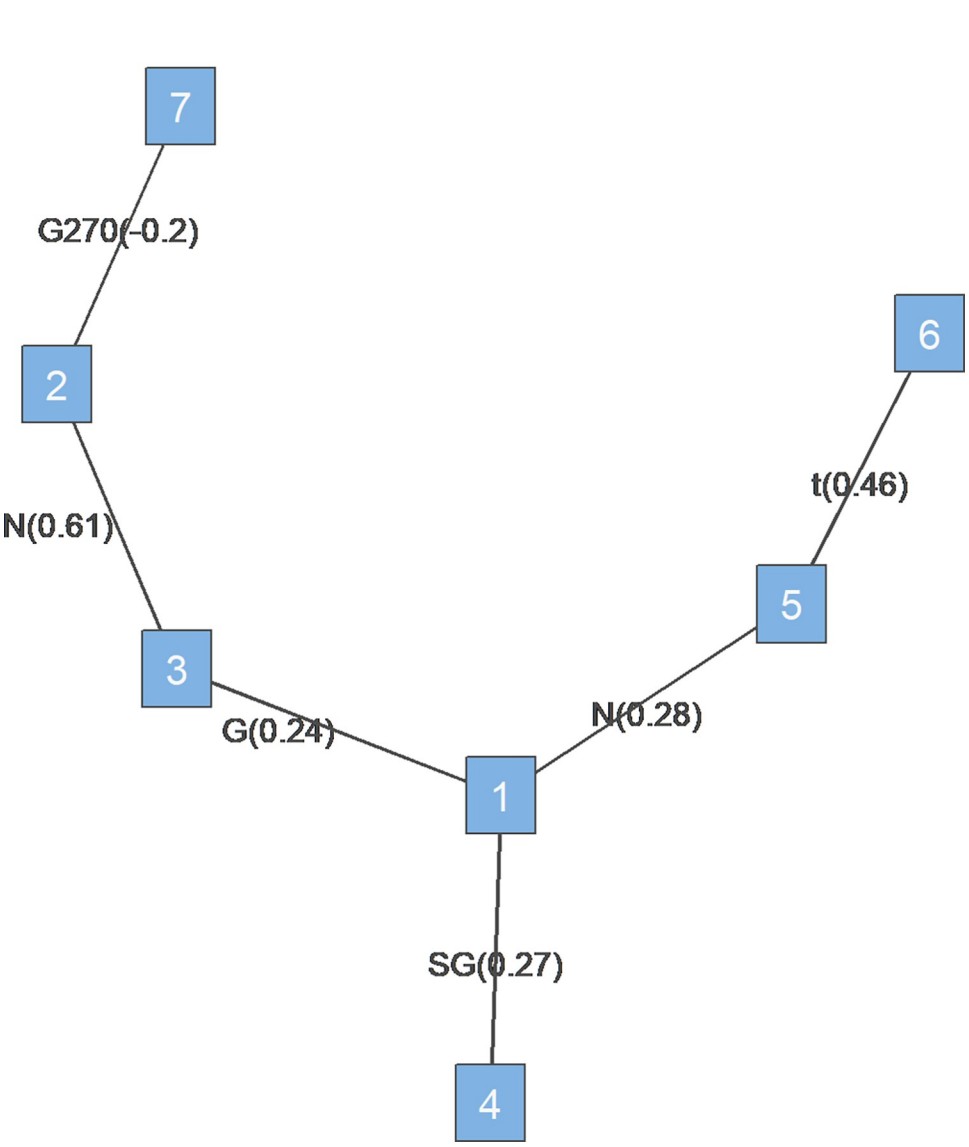

**Fig 1. Tree1 during the Russia-Ukraine war.** Note: 1 to 7 represent "OIL," "AU," "AG," "BEAN," "CORN," "WHEAT," and "S&P500," respectively.

volatility, which is shown in Fig 1. The war-induced energy supply disruptions have not only affected international agricultural market prices but also led to spillover effects on precious metal markets through energy price fluctuations. This transmission mechanism makes markets more vulnerable, reflecting the high sensitivity of other commodities to energy price volatility.

In conclusion, during the Russia-Ukraine war, the dependency structures and price co-movements between commodity and stock markets changed significantly, especially manifested in the inverse relationship between gold and the stock market and the pivotal role of the crude oil market. The heightened market uncertainty due to the war led to a stronger co-movement between gold and silver as safe-haven assets, while the volatility of the oil market had a more pronounced impact on agricultural and precious metal markets. For investors, this study provides insights into asset allocation strategies in the context of geopolitical conflicts: in

**Table 6. Estimated results for tree 1 during COVID-19.**

| tree | edge | family | cop | par | par2 | tau | utd | ltd |
|------|------|--------|-----|-----|------|-----|-----|-----|
| 1 | 7,3 | 1 | N | 0.194 | 0.000 | 0.124 | 0.000 | 0.000 |
| | 7,1 | 2 | t | 0.327 | 10.000 | 0.212 | 0.038 | 0.038 |
| | 1,5 | 4 | G | 1.134 | 0.000 | 0.118 | 0.158 | 0.000 |
| | 5,6 | 14 | SG | 1.224 | 0.000 | 0.183 | 0.000 | 0.239 |
| | 5,4 | 2 | t | 0.609 | 6.192 | 0.417 | 0.227 | 0.227 |
| | 3,2 | 2 | t | 0.766 | 5.138 | 0.555 | 0.401 | 0.401 |

Note: 1 to 7 represent "OIL," "AU," "AG," "BEAN," "CORN," "WHEAT," and "S&P500," respectively.

extreme events such as wars, gold, and oil may serve as effective hedging assets and indicators of risk contagion, helping investors achieve risk diversification and optimize asset allocation in turbulent market conditions.

## R-Vine copula model in the control group

**R-Vine copula model during COVID-19.** From the estimated results in Table 6, we observe several key insights regarding the dependency structure between commodity and stock markets during the COVID-19 period.

Firstly, all commodity markets exhibit positive dependency with the stock market, with Kendall's tau rank correlation coefficients ranging from 0.118 to 0.555. This positive dependency indicates a homogeneous spillover of volatility across all commodity and stock markets. Among them, the silver and gold markets show the strongest correlation, with a Kendall's tau of 0.555, followed by the soybean and corn markets, with a tau of 0.417. This highlights frequent interactions within the precious metals (gold and silver) and agricultural (soybean and corn) markets, indicating a stronger intra-market dependency within these categories and a higher likelihood of homogeneous volatility transmission.

Secondly, we find distinct patterns among different asset pairs. The dependency between crude oil and the S&P500, as well as between gold and silver, follows a t-Copula model, displaying characteristics of thick-tailed correlation. The upper tail dependency (utd) and lower tail dependency (ltd) for these pairs are symmetric, indicating a balanced tail dependence structure. On the other hand, the dependency between corn and wheat is captured by a 90-degree rotated Gumbel Copula (SG), which exhibits lower tail dependence. This suggests a greater probability of downward fluctuation transmission between corn and wheat markets. Other dependencies are represented by a Gaussian copula (N), reflecting independence in tail changes across these assets.

Lastly, Fig 2 illustrates that, unlike during the Russia-Ukraine war, corn stood at the center of volatility spillover among commodity and equity markets. This central positioning implies that corn was a primary conduit for volatility transmission across these markets, with its price fluctuations potentially affecting other commodity markets and the stock market in the same direction. Furthermore, corn's direct links with crude oil, wheat, and soybeans underscore the heightened volatility correlation between agricultural markets and the crude oil market during this period. This interconnectedness reflects the intensified transmission of risk and volatility across these sectors amidst the pandemic.

In summary, the COVID-19 period saw a robust co-movement between commodities and the stock market, with notable tail dependencies in specific pairs such as crude oil with S&P500 and gold with silver. Corn emerged as a central node, linking agricultural and energy markets, signaling an enhanced risk spillover pathway in the global market structure.

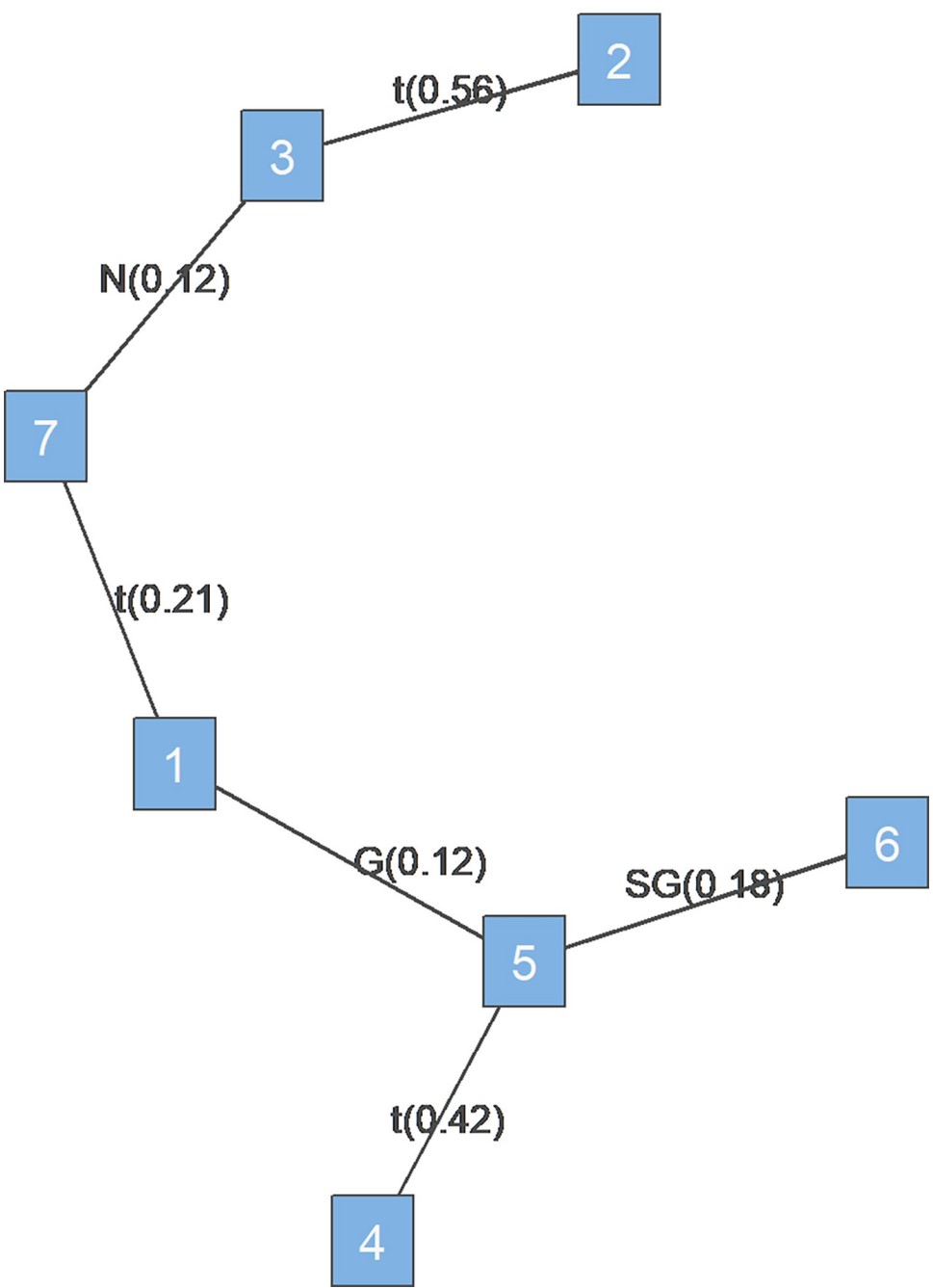

**Fig 2. Tree1 during COVID-19.** Note: 1 to 7 represent "OIL," "AU," "AG," "BEAN," "CORN," "WHEAT," and "S&P500," respectively.

**R-Vine copula during the period of no extreme events.** Finally, in this paper, seven series from 1st January 2014 to 31st December 2018 are selected as the period of no extreme events, and the R-Vine copula model is constructed, whose optimal estimation results in the D-vine structure according to the LL maximum and AIC minimum principles, with the

**Table 7. Control group R-vine copula tree1 estimation results.**

| tree | Edge | family | cop | par | par2 | tau | utd | ltd |
|------|------|--------|-----|------|-------|--------|-------|-------|
| 1 | 6,5 | 2 | t | 0.603 | 7.026 | 0.412 | 0.196 | 0.196 |
| | 5,4 | 2 | t | 0.557 | 7.423 | 0.376 | 0.158 | 0.158 |
| | 4,3 | 1 | N | 0.197 | 0.000 | 0.126 | 0.000 | 0.000 |
| | 3,2 | 2 | t | 0.805 | 9.896 | 0.596 | 0.302 | 0.302 |
| | 2,7 | 2 | t | -0.241 | 10.000 | -0.155 | 0.001 | 0.001 |
| | 7,1 | 2 | t | 0.257 | 10.000 | 0.166 | 0.027 | 0.027 |

Note: 1 to 7 represent "OIL," "AU," "AG," "BEAN," "CORN," "WHEAT," and "S&P500," respectively.

parameters as shown in Table 7. Considering the length, the first-level tree, Tree 1, is still discussed here. From Table 7, it can be seen that the dependence structure during the "no extreme events" control group differs from that during the Russia-Ukraine war.

Firstly, the degree of interdependence among various markets changes. The positive correlation coefficient of Kendall's tau rank for the control group without extreme events is generally lower than during the Russia-Ukraine war. That is, the Russia-Ukraine war enhanced the correlation among markets. The correlation coefficient of Kendall's tau for gold and stocks during the period of no extreme risk is also negative compared to the period of the Russia-Ukraine war, but the absolute value is smaller. The negative correlation between gold and stocks during the period of no extreme risk is less, i.e., the negative transmission of price fluctuations is less likely. This suggests that the hedging effect of asset allocation on risk is stronger in periods of the Russia-Ukraine war if the assets are allocated to the gold and stock markets than in periods without extreme events.

Secondly, the characteristics of the copula distribution in each market have also changed. The pair copula between each commodity and equity market during the COVID-19 epidemic period and the Russia-Ukraine war includes Gaussian copula (N), t copula, Gumbel copula, and surviving Gumbel copula; while it is mostly t copula within the control group, which means that in the absence of exogenous shocks such as epidemics and wars This implies that in the absence of exogenous shocks such as epidemics and wars, there is a symmetrical tail-dependent distribution between the international commodity market and the stock market.

Finally, the dependency structure between markets has also changed. The market dependence structure during the Russia-Ukraine war was an R-Vine structure with the crude oil market as the hub, and the market dependence structure during the pre-war COVID-19 pandemic was an R-Vine structure with the corn market as the hub. During the period without extreme events, the commodity and stock markets were chained in a D-vine structure. Looking at Fig 3, it can be seen that in the control group without extreme events, gold, silver, and soybeans are located in the middle of the chain, which means that these three markets are the most correlated with the other four markets and their price fluctuations are more likely to affect the prices of the other markets. Agricultural products and precious metals were key in connecting the commodity and equity markets in the pre-war period without extreme events.

## Conclusion

In this paper, we use ARMA-GARCH-std to construct a Vine Copula model to study the dependence structure of international commodity and stock markets during the Russia-Ukraine war. In the sample selection, we select seven sets of series, namely, WTI crude oil futures settlement price OIL, London gold closing price AU, London silver closing price AG, CBOT soybean futures closing price BEAN, CBOT corn futures closing price CORN, CBOT

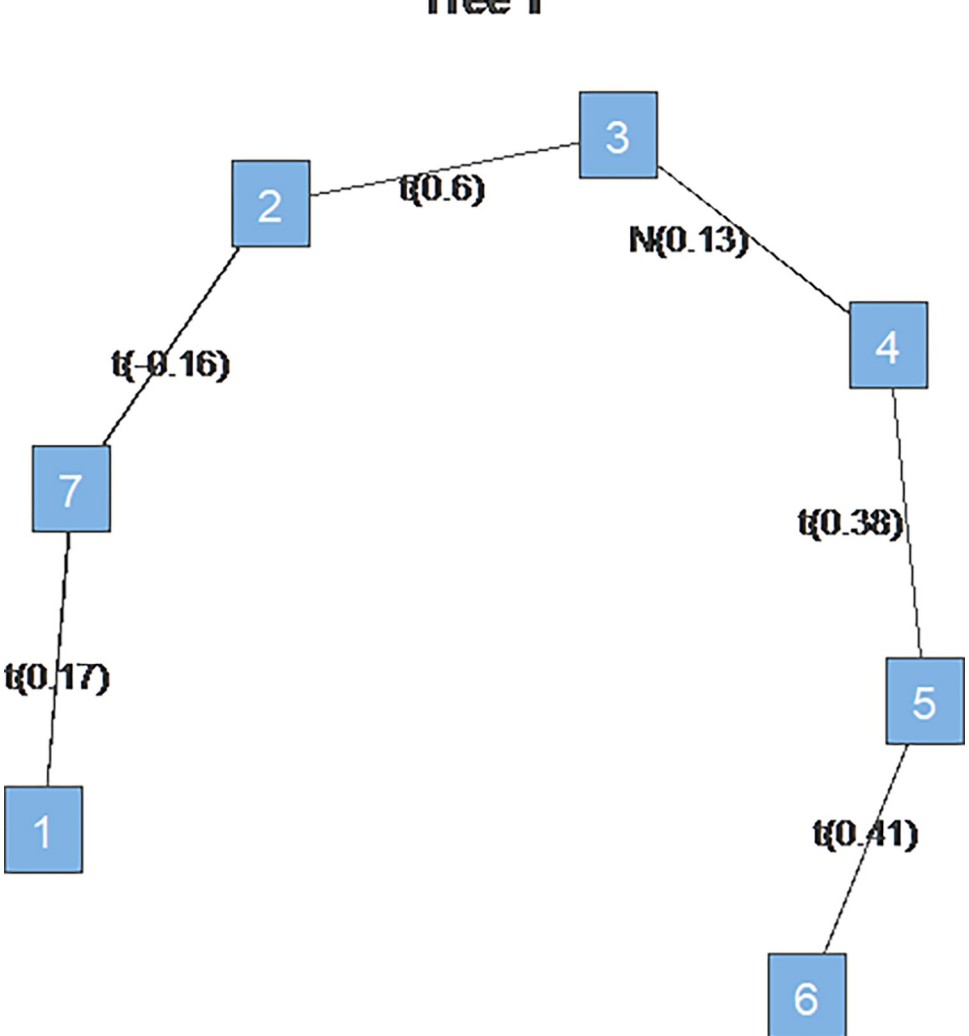

**Fig 3. Tree1 for the control group.** Note: 1 to 7 represent "OIL," "AU," "AG," "BEAN," "CORN," "WHEAT," and "S&P500," respectively.

wheat futures closing price WHEAT, and S&P 500 index. We set the sample period from 24$^{th}$ February 2022 to 1$^{st}$ June 2022 based on the period of the Russia-Ukraine war. After removing the treatment of non-common trading day samples, this paper derives seven sets of differential one-period daily return series for the two sample periods.

We select the optimal order of ARMA based on the above 7 sets of time series according to the AIC information criterion and construct ARMA(m,n)-GARCH(1,1) models as their marginal distribution functions for each series, respectively, where m and n are the optimal lagged orders of ARMA, and the GARCH types include sGARCH, APARCH, and GJR-GARCH. By the KS test, the fitted standard residuals of each time series satisfy the condition of uniform distribution, and the vine copula can be constructed. We construct the R vine copula structure on this basis and derive the research results of the phase-dependent structure. As a comparison, this paper selects samples during the COVID-19 pandemic and periods without extreme events, and the vine copula structure is also constructed.

Through the empirical study, the main findings of this paper are as follows. During the Russia-Ukraine war, the structure of the vine connecting each market was still R-Vine, but observing its structure and correlation coefficient Kendall's tau shows that during this period, the crude oil futures OIL serves as a connecting hub between the markets, and the OIL is directly linked to the soybean market BEAN and corn market CORN, representing agricultural commodities, as well as the silver market AG for precious metals. This suggests that price volatility in the crude oil market may be contagious in the same direction as in the agricultural and precious metals markets. A look at the correlation coefficient tau shows that the stock market S&P500 and the gold market AU are negatively correlated, which indicates the possibility of reverse price volatility transmission between these two markets, further suggesting that asset allocation to the equity market and the gold market may play a hedging role for asset risk. In addition, looking at the correlation coefficients between the rest of the markets, we can see that the rest of the tau values are positive and increased compared to the period of the COVID-19 epidemic, and the markets are more closely correlated. Overall, the vine copula structure during the Russia-Ukraine war shows, first of all, that crude oil became the key commodity behind the war. This finding also further reveals the importance of crude oil in the war. The war has increased policy uncertainty in the international community, which has contributed to price volatility risks in commodity and stock markets, and Russia's energy supply contraction has further exacerbated the propagation of crude-driven price volatility risks in international markets. Secondly, the war between the two countries affected the international supply of agricultural products, energy price volatility is contagious to international agricultural markets, and Ukraine, an important exporter of agricultural products, has seen a contraction in production and exports of all types of agricultural products. The war could not only further amplify the spread of price risks in financial and commodity markets but also exacerbate the international food security problems that already existed during COVID-19. Lastly, crude oil price fluctuations were also homogeneously transmitted to the precious metals market, while allocating assets in the gold and stock markets had the potential to diversify asset risk.

Finally, to further observe changes in the dependency structure, we introduce the pre-war COVID-19 epidemic sample and the sample without extreme events. During the COVID-19 epidemic, the vine structure connecting each market is the R-Vine structure, and observing tau rank correlation coefficients and tree1 plots for tree1 between each market shows that the corn futures market is the hub connecting the other commodity markets and the stock market. The correlation coefficients between each market are positive, with mostly smaller tau values than during the Russia-Ukraine war, indicating the possibility of homogeneous transmission of price fluctuations across markets but with less dependence than during the Russia-Ukraine war. The largest tau values are found between the gold AU and silver AG markets and between the soybean BEAN and corn CORN markets, i.e., the dependence is strong, and the possibility of homogeneous transmission of fluctuations is high. This suggests that food security issues became a central international concern during the COVID-19 epidemic and that volatility in agricultural prices may be transmitted homogeneously to other commodities and stock markets. While the structure of vine copula in the comparison period without extreme events is a chain-like D-vine, and although it does not have a market with the presence of a key pivotal position like the vine structure during the Russia-Ukraine war, the gold (AU), silver (AG), and soybean (BEAN) markets are more correlated with other markets in the comparison period, precious metals and agricultural products dominate the overall structure. In addition, tau values across markets during periods without extreme events are generally lower than tau values during the Russia-Ukraine war, with lower dependence. This further suggests that the epidemic and war shocks have enhanced the interdependence of international commodity and

equity markets and that the financial markets are increasingly linked to each other. Overall, based on the comparative results, we find that the Russia-Ukraine war has increased the dependence on the commodity and stock markets, and the core of the dependence structure has shifted from agricultural products and precious metals to crude oil.

## Author Contributions

**Conceptualization:** Cheng Zhang, Shuo Liu.

**Formal analysis:** Cheng Zhang, Shuo Liu.

**Investigation:** Cheng Zhang, Shuo Liu, Mimi Qin.

**Methodology:** Cheng Zhang, Shuo Liu, Bin Gao.

**Project administration:** Cheng Zhang, Bin Gao.

**Supervision:** Mimi Qin, Bin Gao.

**Validation:** Mimi Qin, Bin Gao.

**Visualization:** Mimi Qin.

**Writing – original draft:** Cheng Zhang, Shuo Liu.

**Writing – review & editing:** Mimi Qin, Bin Gao.

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
