## [Decision Letter · Decision Letter 0]

25 Oct 2024

PONE-D-24-41457The dependency structure of international commodity and stock markets after the Russia-Ukraine warPLOS ONE

Dear Dr. Gao,

Thank you for submitting your manuscript to PLOS ONE. After careful consideration, we feel that it has merit but does not fully meet PLOS ONE’s publication criteria as it currently stands. Therefore, we invite you to submit a revised version of the manuscript that addresses the points raised during the review process.

The article, in its current state, is not publishable on PLOS ONE. I encourage the authors to review the suggestions provided by the reviewers and follow them carefully to improve the article. Based on the reviewers' comments, my decision is Major Revisions.

We look forward to receiving your revised manuscript.

Kind regards,

Alessandro Mazzoccoli, Ph.D.

Academic Editor

PLOS ONE

Journal Requirements: When submitting your revision, we need you to address these additional requirements. 1. Please ensure that your manuscript meets PLOS ONE's style requirements, including those for file naming. The PLOS ONE style templates can be found at https://journals.plos.org/plosone/s/file?id=wjVg/PLOSOne_formatting_sample_main_body.pdf and https://journals.plos.org/plosone/s/file?id=ba62/PLOSOne_formatting_sample_title_authors_affiliations.pdf 2. Thank you for stating the following financial disclosure: "‘Guangxi One Thousand Young and Middle-Aged College and University Backbone Teachers Cultivation Program’ Humanities and social sciences projects (2020QGRW016)." Please state what role the funders took in the study.  If the funders had no role, please state: ""The funders had no role in study design, data collection and analysis, decision to publish, or preparation of the manuscript.""  If this statement is not correct you must amend it as needed. Please include this amended Role of Funder statement in your cover letter; we will change the online submission form on your behalf. 3. We note you have included a table to which you do not refer in the text of your manuscript. Please ensure that you refer to Table 1 and 7 in your text; if accepted, production will need this reference to link the reader to the Table.

Reviewers' comments:

Reviewer's Responses to Questions

**Comments to the Author**

1. Is the manuscript technically sound, and do the data support the conclusions?

Reviewer #1: Yes

Reviewer #2: Partly

2. Has the statistical analysis been performed appropriately and rigorously? 

Reviewer #1: Yes

Reviewer #2: No

3. Have the authors made all data underlying the findings in their manuscript fully available?

Reviewer #1: No

Reviewer #2: Yes

4. Is the manuscript presented in an intelligible fashion and written in standard English?

Reviewer #1: Yes

Reviewer #2: No

5. Review Comments to the Author

Reviewer #1: The authors perform a thorough and statistically sound analysis of the dependency structure between a number of commodities and the S&P 500 index, and of its changes during different time periods. Interestingly, they compare a time of market stability with two different periods of extreme events (caused by the pandemic and by war) and find evidence of a shift between the core of the dependence structure between stock market and commodities. The methodology of choice is well-justified by tests on the properties of the time series under study. It is also perfectly suitable for investigating the research question of the paper. The process of data cleaning, often neglected, here is presented in succinct but clear detail. The authors find interesting results, which are duly highlighted and presented clearly, and which can be useful for investors and market players at large.

The only weaknesses to highlight are not methodological but cosmetic. They are the following:

a. The presence of website links inside the text (as on page 3) should be avoided: consider using footnotes or references instead

b. The commodities corresponding to each number in the edges of tables 5-7 and in figures 1-3 should be made explicit

c. The resolution of figures 1-3 is low. If possible, it should be increased

d. A few typos, slight language errors and repetitions appear. Here are some:

line 61 “exogenous backgrounds”, line 63 “outbroke”, line 75 “expos-es”, line 109 “MR”, line 110 “corresponding to the corresponding”, the sentence on lines 150-152 should be made clearer, line 220 “the st”, line 288 “tree treetree1”, a period is missing after “here” and “The” should not be capitalized.

Reviewer #2: The paper has an interesting motivation and a valid research question. The work aims to investigate the changes in the dependence structure between several commodity markets and the stock market before and during the Russia-Ukraine war. Methodologically, they estimate an ARMA-GARCH model to the data and apply an R-vine copula model to the resulting standardized residuals. Their findings show that the outbreak of the war has increased the interdependence among the commodities and stock markets, with crude oil as key commodity.

Despite the engaging motivation behind the paper, the implementation still needs much work from different perspectives.

GENERAL WRITING

Overall, the paper is poorly written, both in vocabulary (i.e., too many repetitions and ambiguous use of terms) and in the structure of the paragraphs. It is hard to understand the motivation driving this research (one can only infer it), while it should be clear in the abstract and the introduction; the methodology is hardly described and, when hinted at, it is unclear and difficult to grasp, the reader can only understand it once she gets to the "Methodology" section, whereas it should figure clearly in the introduction as well.

The title suggests that the pivotal event of the analysis is the Russia-Ukraine war, however, the introduction focuses almost entirely on the COVID-19 pandemic, which is another event of major disruption for financial markets. This might create confusion upfront on what is the true goal of the research.

The technical terminology is often inappropriate. For instance, the authors use coding terms that are used in R but are not suited to the text: the term "ARMA-GARCH-std", where the "std" is generally used in R to denote the Student t distribution, is not elaborated, nor is there any hint at its interpretation. Another example is the use of the term "simulation" in the context of the ARMA-GARCH, how is it a simulation when real data are used to estimate the parameters? Moreover, the terms "training group" and "control group" are not elaborated nor appropriately contextualized.

Other minor inaccuracies are related to unexplained acronyms, such as AIC, BIC, KS.

BIBLIOGRAPHY

Overall, the bibliography is non-exhaustive and the cited references are mostly either unrelated to the topic of discussion or missing where needed (e.g., line 88 in the manuscript).

In the text, the citing system is not uniform, sometimes the cited papers appear by full name and year and other times by the reference number in square brackets in the "References" section.

EMPIRICAL ANALYSIS

While the use of R-vine copulas to address changes in the dependence structure is new in the context of the Russia-Ukraine war, the empirical analysis in the ARMA-GARCH estimation is not precise in its description.

1. The data source is not indicated.

2. What justifies the use of different GARCH models? Is there an interest in capturing asymmetry? Was this tested (e.g., using a negative size bias test or similar)? Also, neither of the models is explained.

3. The AIC criterion alone is not sufficient for model selection, but Table 3 and Table 4 do not report any other additional selection criteria, e.g., BIC, Hannan-Quinn (HQ).

4. At the beginning of the empirical analysis, the estimation seems to be performed under the assumption of Gaussian distribution, but then the parameters in Table 4 include the shape coefficient which is typically related to the Student t; similarly, the caption of the same table reads "ARMA-GARCH-std". It is inferred that it is a consequence of the results of the Jarque-Bera test for normality, but this step needs further elaboration.

5. Table 3 does not report the significance of the estimated coefficients.

OTHER REMARKS

The results are poorly discussed. There is barely any critical thinking behind the comments on the results and there are no considerations of the systemic implications of the analysis. How are the results informative for the reader and how do they answer the research question? This is not clear enough and ambiguous. Overall, the idea is valid, and the methodology per se is sound but the execution is chaotic and often inaccurate, or at least inaccurately explained.

6. PLOS authors have the option to publish the peer review history of their article (what does this mean?). If published, this will include your full peer review and any attached files.

Reviewer #1: No

Reviewer #2: No

---

## [Author Response · Author response to Decision Letter 0]

22 Nov 2024

Dear Editor:

Re: PONE-D-24-41457 titled “The dependency structure of international commodity and stock markets after the Russia-Ukraine war”

We are very grateful to your and the reviewers’ critical comments and thoughtful suggestions. Based on these comments and suggestions, we have made careful modification on the original manuscript. All deletions from the text are marked in red in the revised manuscript, while additions and modifications are highlighted in blue, making them easy to identify. Some of your questions were answered below.

Once again, we acknowledge your comments and constructive suggestions very much, which are valuable in improving the quality of our manuscript. 

Here are our responses to the reviewers’ comments one-by-one. 

Reviewer #1:

Comment1: 

The authors perform a thorough and statistically sound analysis of the dependency structure between a number of commodities and the S&P 500 index, and of its changes during different time periods. Interestingly, they compare a time of market stability with two different periods of extreme events (caused by the pandemic and by war) and find evidence of Page 5 a shift between the core of the dependence structure between stock market and commodities. The methodology of choice is well-justified by tests on the properties of the time series under study. It is also perfectly suitable for investigating the research question of the paper. The process of data cleaning, often neglected, here is presented in succinct but clear detail. The authors find interesting results, which are duly highlighted and presented clearly, and which can be useful for investors and market players at large. The only weaknesses to highlight are not methodological but cosmetic. They are the following: a. The presence of website links inside the text (as on page 3) should be avoided: consider using footnotes or references instead b. The commodities corresponding to each number in the edges of tables 5-7 and in figures 1-3 should be made explicit c. The resolution of figures 1-3 is low. If possible, it should be increased d. A few typos, slight language errors and repetitions appear. Here are some: line 61 “exogenous backgrounds”, line 63 “outbroke”, line 75 “expos-es”, line 109 “MR”, line 110 “corresponding to the corresponding”, the sentence on lines 150-152 should be made clearer, line 220 “the st”, line 288 “tree treetree1”, a period is missing after “here” and “The” should not be capitalized.

Response1:

Thank you for acknowledging our thorough and statistically sound analysis of the dependency structure between commodities and the S&P 500 index across different time periods. We appreciate your recognition of our research methodology as suitable for addressing the research questions of this paper, and we are pleased to hear that the data cleaning process was clearly presented.

1. Regarding the issue of website links in the text: 

Response: We have removed all website links from the main text and replaced them with footnotes to enhance the academic rigor of the paper.

2. Concerning the clarity of commodities in tables and figures: 

Response: We have explicitly labeled each number corresponding to commodities in notes below Tables 5-7 and Figures 1-3, allowing readers to better understand the data. Note: 1 to 7 represent "OIL," "AU," "AG," "BEAN," "CORN," "WHEAT," and "S&P500," respectively.

3. Regarding the resolution of figures:

 Response: We have enlarged the size of Figures 1-3 and improved their resolution to ensure that the information in the charts is clearly visible.

4. Concerning typos and language errors in the text:

 Response: We have carefully proofread the entire manuscript and corrected all typos and language errors, including those specifically pointed out by the reviewer. We have thoroughly reviewed the text to ensure accuracy and fluency in language.

Examples of specific changes made:

• Line 61: "Exogenous backgrounds." We believe that not only was the term "exogenous backgrounds" used inappropriately, but the sentence was also unclear. The original sentence was: "However, commodity and financial market correlations based on exogenous backgrounds such as the COVID-19 outbreak and the Russia-Ukraine war remain relatively rare and have increased correlation between financial and commodity markets." It has now been changed to: "While there has been existing research on the relationship between commodity and financial markets under external shocks, studies on the deep structural changes in dependencies induced by the Russia-Ukraine war are still limited."

• Line 63: "outbroke" has been corrected to "emerged." The original sentence was: "Among the international exogenous events of recent years, COVID-19, which outbroke at the end of 2019, has become a global public health crisis, and its pandemic continues to increase economic uncertainty and cause macroeconomic depression." We have modified it to: "Baker et al. [2] highlight that in the current context of highly interconnected global financial and commodity markets, exogenous shocks such as the COVID-19 pandemic, which emerged at the end of 2019, significantly exacerbate market volatility. This outbreak not only precipitated a global public health crisis but also resulted in heightened economic uncertainty and a macroeconomic recession."

• Line 75: "expos-es". The original sentence corresponding to line 75 was: "This expos-es the possibility that the Russia-Ukraine war could affect international stocks and spill over into commodity markets, changing the existing relationship between international financial and commodity markets. Referring to studies on the impact of previous wars on financial markets." We have changed it to: "This suggests that the Russia-Ukraine war may affect international stock markets and spill over into commodity markets, thereby altering the existing relationship between international financial markets and commodity markets."

• Line 109: "MR" has been corrected to "MA". We apologize for our carelessness; this was a typographical error, and it actually refers to the "MA model," which is the moving average model.

• Line 110: "corresponding to the corresponding" has been rewritten as "corresponding to the respective variables."

• Lines 150-152: We have reconstructed this sentence to improve its clarity. The original sentence was: "On this basis, Joe (1996) proposed pair-copula construction (PPC) [21], while Bedford and Cooke (2001) proposed R-Vine copula construction based on PPC [22], which can decompose the multivariate density function into the marginal density and a series of unconditional or conditional pair-copula, thus capturing the dependence and degree of dependence between every two variables in n-dimensional variables. The R-vine copula is defined as follows:" It has now been modified to: "Building on this, Joe [16] proposed the Pair-Copula Construction (PPC). Subsequently, Bedford and Cooke [17] developed the R-Vine Copula construction based on PPC. The R-Vine Copula can decompose a multivariate density function into marginal densities and a series of unconditional or conditional pair-Copulas, thereby capturing the dependence and degree of dependence between any two variables in an n-dimensional space. The definition of the R-Vine Copula is as follows:"

• Line 220: "the st". The original sentence was: "The results of KS the st could not reject the original hypothesis, that is, each series after transformation obeyed uniform distribution and satisfied the conditions for the construction of Vine-copula." We apologize for the typographical error here and have modified this sentence to: "The results of the KS test failed to reject the null hypothesis, suggesting that the transformed series follow a uniform distribution, thereby meeting the conditions required for constructing a Vine copula model."

• Line 288: "tree treetree1" has been corrected to "Tree1". The missing period after "here" has been added, and the capitalization of "The" has also been corrected to lowercase. Specifically, the original sentence was: "Considering the space, the first level tree treetree1 is still discussed here. From Table 6, it can be seen that compared to the COVID-19 epidemic. The dependence structure of each commodity market and the stock market during the control period differs from that during the Russia-Ukraine war." It has been modified to: "Considering the length, the first-level tree, Tree 1, is still discussed here. From Table 6, it can be seen that the dependence structure during the 'no extreme events' control group differs from that during the Russia-Ukraine war."

Reviewer #2:

Comment1: GENERAL WRITING 

Overall, the paper is poorly written, both in vocabulary (i.e., too many repetitions and ambiguous use of terms) and in the structure of the paragraphs. It is hard to understand the motivation driving this research (one can only infer it), while it should be clear in the abstract and the introduction; the methodology is hardly described and, when hinted at, it is unclear and difficult to grasp, the reader can only understand it once she gets to the "Methodology" section, whereas it should figure clearly in the introduction as well. The title suggests that the pivotal event of the analysis is the Russia-Ukraine war, however, the introduction focuses almost entirely on the COVID-19 pandemic, which is another event of major disruption for financial markets. This might create confusion upfront on what is the true goal of the research. The technical terminology is often inappropriate. For instance, the authors use coding terms that are used in R but are not suited to the text: the term "ARMA-GARCH-std", where the "std" is generally used in R to denote the Student t distribution, is not elaborated, nor is there any hint at its interpretation. Another example is the use of the term "simulation" in the context of the ARMA-GARCH, how is it a simulation when real data are used to estimate the parameters? Moreover, the terms "training group" and "control group" are not elaborated nor appropriately contextualized. Other minor inaccuracies are related to unexplained acronyms, such as AIC, BIC, KS.

Response1:

Thank you for your review and valuable feedback on our manuscript. We have carefully considered your comments and made the following revisions and improvements in response to the specific issues you raised.

1. Issues with writing quality and terminology: 

Response: We acknowledge the shortcomings in the writing quality and use of terminology in the paper. To enhance the clarity and professionalism of the paper, we have thoroughly proofread and rewritten the entire text, particularly revising and clarifying repetitive and ambiguous terms. We have ensured that each term is clearly defined at its first occurrence and used consistently throughout the paper. As you pointed out, the term "simulation" is not appropriate for our study, and we have replaced it with more suitable terms, such as "estimated." Furthermore, we appreciate your pointing out the improper use of technical terms. We have re-examined and corrected the use of these terms to ensure their applicability and accuracy in the text. In particular, we have modified "ARMA-GARCH-std" to clarify that "std" refers to the Student's t-distribution and have made sure that all technical terms are properly contextualized and explained.

2. Unclear research motivation: 

Response: We understand the importance of clear research motivation and have more explicitly articulated the driving factors of the study in the abstract and introduction. We have reorganized the introduction to more directly highlight the significance of the Russia-Ukraine war to the study while reducing the discussion of the COVID-19 pandemic to avoid confusion. Here is our revised introduction:

In recent decades, the deepening integration of global trade has provided investors in financial markets with a broader and more complex range of investment choices and portfolio strategies, while the trend of commodity financialization has also intensified [1]. The link between international commodity markets and financial markets, represented by stock markets, has become increasingly closer, and the price fluctuations of assets across these markets are often interrelated. Baker et al. [2] highlight that in the current context of highly interconnected global financial and commodity markets, exogenous shocks such as the COVID-19 pandemic, which emerged at the end of 2019, significantly exacerbate market volatility. This outbreak not only precipitated a global public health crisis but also resulted in heightened economic uncertainty and a macroeconomic recession. Subsequently, the Russo-Ukrainian War, which began on February 24, 2022, introduced further substantial uncertainty into the global policy environment, exacerbating volatility in international financial markets and potentially having enduring effects on the global economy [3, 4]. Policy uncertainty alters investment decisions and consumer behavior, thereby affecting financial market returns and introducing spillover effects that increase market volatility [5]. Albulescu et al. [6] noted that policy uncertainty amplifies the potential for risk transmission between commodity markets and financial markets. As a result, studies investigating the relationship between commodity markets and financial markets under external shocks have emerged, covering markets such as oil, gold, stock markets, and exchange rates [7, 8]. Brune et al. [9] specifically pointed out that the sudden outbreak of war increases market uncertainty, and international financial markets often react negatively to war and armed conflict events, which is reflected in falling stock prices.

The Russia-Ukraine war, as a sudden geopolitical conflict, brought new policy uncertainty to the international community [10]. The New York Times reported that on the day the war broke out, the S&P 500 index fell by more than 10%[11]. This suggests that the Russia-Ukraine war may affect international stock markets and spill over into commodity markets, thereby altering the existing relationship between international financial markets and commodity markets. The Russia-Ukraine war not only increased the uncertainty surrounding business operations but also had a significant negative impact on global stock market returns. Although the war occurred primarily between Russia and Ukraine, economic globalization magnified its impact, causing ripple effects on the neighboring countries and the UN member states calling for a ceasefire[10]. In the commodity market, the war is expected to have a particularly pronounced impact due to Russia's key role in energy supply and Ukraine's importance in agricultural supply, with oil being a critical commodity in international markets [12].

With the deepening of global economic integration, the interdependence between international financial markets and commodity markets has grown, meaning that price fluctuations and risks in one asset market may spill over to other markets. Analyzing market dependencies during major external shocks helps investors better allocate assets. While there has been existing research on the relationship between commodity and financial markets under external shocks, studies on the deep structural changes in dependencies induced by the Russia-Ukraine war are still limited. This study aims to analyze the relationship between international commodity markets and stock markets under the external shock of the Russia-Ukraine war. We selected the prices of six commodity futures (oil, gold, silver, soybean, corn, and wheat) and the S&P 500 index, using ARMA-GARCH models to fit the marginal distributions of each market and employing the R-Vine Copula structure to analyze the dependencies between these marginal distributions. To ensure the robustness of the data, we excluded irregular trading days and constructed independent R-Vine Copula structures for the periods of the Russia-Ukraine war, the pre-war COVID-19 pandemic, and periods without major external shocks. This analysis aims to better understand the impact of the Russia-Ukraine war on the dependence structure between commodity and stock markets.

By comparing the dependency structures between commodity and stock market

---

## [Decision Letter · Decision Letter 1]

10 Dec 2024

The dependency structure of international commodity and stock markets after the Russia-Ukraine war

PONE-D-24-41457R1

Dear Dr. Gao,

We’re pleased to inform you that your manuscript has been judged scientifically suitable for publication and will be formally accepted for publication once it meets all outstanding technical requirements.

Kind regards,

Alessandro Mazzoccoli, Ph.D.

Academic Editor

PLOS ONE

Additional Editor Comments (optional):

Reviewers' comments:

Reviewer's Responses to Questions

**Comments to the Author**

1. If the authors have adequately addressed your comments raised in a previous round of review and you feel that this manuscript is now acceptable for publication, you may indicate that here to bypass the “Comments to the Author” section, enter your conflict of interest statement in the “Confidential to Editor” section, and submit your "Accept" recommendation.

Reviewer #1: All comments have been addressed

Reviewer #2: All comments have been addressed

2. Is the manuscript technically sound, and do the data support the conclusions?

Reviewer #1: Yes

Reviewer #2: Yes

3. Has the statistical analysis been performed appropriately and rigorously? 

Reviewer #1: Yes

Reviewer #2: Yes

4. Have the authors made all data underlying the findings in their manuscript fully available?

Reviewer #1: Yes

Reviewer #2: Yes

5. Is the manuscript presented in an intelligible fashion and written in standard English?

Reviewer #1: Yes

Reviewer #2: Yes

6. Review Comments to the Author

Reviewer #1: The new version of the manuscript addresses in detail all previous concerns. There are marked improvements in the clarity of the text and, in general, of the analysis.

Reviewer #2: The comments from the previous review process have been successfully addressed. I only highlight the following minor comments:

1. A small typo in equation (1) where there is a misplaced "a" in the ARMA(p,q) model.

2. The references should be listed in alphabetical order.

7. PLOS authors have the option to publish the peer review history of their article (what does this mean?). If published, this will include your full peer review and any attached files.

Reviewer #1: No

Reviewer #2: No

---

## [Editor Report · Acceptance letter]

16 Dec 2024

PONE-D-24-41457R1 

PLOS ONE

Dear Dr. Gao, 

I'm pleased to inform you that your manuscript has been deemed suitable for publication in PLOS ONE. Congratulations! Your manuscript is now being handed over to our production team.

Kind regards, 

on behalf of

Dr. Alessandro Mazzoccoli 

Academic Editor

PLOS ONE